# Revision and Extension of a Generally Applicable Group-Additivity Method for the Calculation of the Standard Heat of Combustion and Formation of Organic Molecules

**DOI:** 10.3390/molecules26206101

**Published:** 2021-10-10

**Authors:** Rudolf Naef, William E. Acree

**Affiliations:** 1Department of Chemistry, University of Basel, 4003 Basel, Switzerland; 2Department of Chemistry, University of North Texas, Denton, TX 76203, USA; acree@unt.edu

**Keywords:** group-additivity method, Gauss–Seidel diagonalization, heat of combustion, heat of formation

## Abstract

The calculation of the heats of combustion ΔH°_c_ and formation ΔH°_f_ of organic molecules at standard conditions is presented using a commonly applicable computer algorithm based on the group-additivity method. This work is a continuation and extension of an earlier publication. The method rests on the complete breakdown of the molecules into their constituting atoms, these being further characterized by their immediate neighbor atoms. The group contributions are calculated by means of a fast Gauss–Seidel fitting calculus using the experimental data of 5030 molecules from literature. The applicability of this method has been tested by a subsequent ten-fold cross-validation procedure, which confirmed the extraordinary accuracy of the prediction of ΔH°_c_ with a correlation coefficient R^2^ and a cross-validated correlation coefficient Q^2^ of 1, a standard deviation σ of 18.12 kJ/mol, a cross-validated standard deviation S of 19.16 kJ/mol, and a mean absolute deviation of 0.4%. The heat of formation ΔH°_f_ has been calculated from ΔH°_c_ using the standard enthalpies of combustion for the elements, yielding a correlation coefficient R^2^ for ΔH°_f_ of 0.9979 and a corresponding standard deviation σ of 18.14 kJ/mol.

## 1. Introduction

The present compilation of data on the heat of combustion and formation of more than 5000 organic molecules and their comparison with theoretical calculations based on a generally applicable atom-groups additivity method is a continuation of theoretical studies on the prediction of various molecular descriptors published in an earlier paper [1]. While this publication primarily focused on the extraordinary versatility of the applied version of the atom-group additivity method for a large number of descriptor predictions, which has been proven by its extension to several further molecular descriptors in subsequent papers [2,3,4,5,6], the present interest rests on the further increase in the trustworthiness of their calculated heats of combustion and formation and their extension to compound classes not yet covered by the earlier paper, particularly the ionic liquids. Previous versions of heat-of-combustion calculations have been based on the additivity of bond energies [7,8,9], on empirical relations within a series of molecules and their heat of combustion [10,11], on the “heat of atomization” [12], on the combustion value of the electrons in a molecule, corrected for its structural and functional features [13,14], or on the “molecular oxygen balance” [15], all of them outlined in more detail in [1]. An indirect approach to the prediction of the heat of combustion—due to their direct interdependence—is via the calculation of the heat of formation of a molecule, which is either accessible through elaborate quantum-theoretical methods (e.g., [16]) or through a group-additivity method [17,18] similar to the present one. Most of these various approaches have been optimized for a certain class of compounds and are therefore not generally applicable. In contrast, the present calculation method is easily extendable and in principle enables the calculation of the heat of combustion and formation of literally any organic molecule under the sun.

## 2. Method

The calculations rest upon a database of at present 34,380 molecules, recorded in their geometry-optimized 3D conformation, encompassing pharmaceuticals, plant protectors, dyes, ionic liquids, liquid crystals, metal-organics, intermediates, and many more, wherein—among many further experimentally determined and calculated molecular descriptors—for 5560 of them, the published experimental combustion and/or formation enthalpies have been stored. In order to avoid structural ambiguity, all six-membered aromatic rings have been defined by six aromatic bonds, in contrast to the more commonly used single-double-bond alternating style. Furthermore, for the same reason, the positive charge in amidinium, pyrazolium, and guanidinium fragments is positioned on the carbon atom between the nitrogen atoms, incidentally in better conformance with the true situation, as shown in, e.g., Figure 1 in [3]. (For the carboxylate or the nitro group, the analogous consideration of charge equilibration is not required within the present atom-group concept, as they are unambiguously defined.) Finally, compounds containing both acidic and basic groups, in particular primary alkylamines (e.g., amino acids) or guanidines (e.g., in creatine or arginine), are treated as zwitter-ionic molecules.

### 2.1. Definition of the Atom Groups

The principle of the breakdown of a molecule into its atom groups in a computer-readable form has been outlined in detail in [1]. Consequently, their naming and meaning are retained in the present work as explained in Table 1 of [1]. However, since then, a number of further atom groups had to be added to the group-contribution parameters set in order to cover the considerable amount of additional, structurally variable molecules. In particular, the inclusion of ordinary salts and ionic liquids required the charged atom groups listed and explained in Table 1, which are interpreted analogously by the computer algorithm as the remaining ones. (Some of these atom groups have already been introduced for the calculation of the liquid viscosity of molecules in [3].)

The atom groups do not take into account the characteristics of the molecules’ three-dimensional structures, such as intramolecular hydrogen-bridge bonds, intramolecular H-H interactions, or ring-strain forces. These effects have summarily been considered by means of the special groups listed and explained in Table 2, wherein the column titles are not to be interpreted literally. With regard to the ring-strain contributions (Angle60, Angle90, and Angle102), caused by forced angle constriction at each ring atom in small rings, it should be stressed that the calculated values inherently also encompass the effect of the compensatory angle widening between the ring atoms and any further atoms attached to them (e.g., the H-C-H and H-C-C angles on cyclopropane). These special groups are treated just like the ordinary atom groups in the calculation of their contribution as well as the subsequent molecular descriptor value.

### 2.2. Calculation of the Group Contributions

The parameter values of the atom and special groups are calculated in four steps, outlined in detail in [1]: the first step creates a temporary compounds list and adds those compounds from the database into it for which the experimental heat of combustion is known. Secondly, for each of the “backbone” atoms (i.e., atoms bound to at least two other direct neighbor atoms) in the molecules, its atom group is defined according to the rules defined in [1], corresponding to the atom type and neighbors’ terms listed in Table 4, and then its occurrence in the molecule is counted. Next, an M × (N + 1) matrix is generated, where M is the number of molecules, where N + 1 is the number of atoms and special groups of Table 4 plus the molecules’ experimental heats of combustion, and where each matrix element (i,j) receives the number of occurrences of the jth atomic or special group in the ith molecule. Finally, normalization of this matrix into an Ax = B matrix and its subsequent balancing using a fast Gauss–Seidel calculus [19] yields the group contributions x, which are shown in Table 4.

### 2.3. Calculation of the Standard Heats of Combustion and Formation

The subsequent calculation of the heat of combustion ΔH°(c) is a simple summing up of the contributions of the atom groups in a molecule using the values shown in Table 4, applying Equation (1), wherein *a_i_* and *b_j_* are the contribution values, *A_i_* is the number of occurrences of the *i*th atom group and *B_j_* is the number of occurrences of the special groups.
∆H^°^_c_ = Σ_i_ a_i*_A_i_ + Σ_j_ b_j*_B_j_(1)

It is immediately evident that these calculations are limited to compounds for which each atom group contained in it (excluding the special groups) has its corresponding one shown in Table 4. Beyond this, in order to receive reliable results, only “valid” group contributions are to be used, i.e., contributions that have been supported in the group-parameters calculation by at least three independent molecules, i.e., by the number in the rightmost column of Table 4 exceeding 2. As a consequence, the statistics data at the bottom of Table 4 show that the number of compounds for which finally the heat of combustion is calculated (lines B, C, and D) is smaller than that on which the computation of the complete set of group contributions is based (line A).

The heat of formation of the molecules is immediately calculated from their heat of combustion by the subtraction of the standard enthalpies of combustion of the elements as given in [20,21].

In Table 3, a simple example may explain the use of Table 4: the experimental heat of combustion of 4-methylene-2-oxetanone (diketene) is −1913.4 kJ/mol [21]. The atom groups and the special group defining this compound are collected in Table 3 and yield a calculated value of −1903.2 kJ/mol.

### 2.4. Cross-Validation Calculations

The results of the heat-of-combustion data are immediately tested for plausibility using a 10-fold cross-validation algorithm, requiring 10 recalculations that guarantee that each of the complete set of compounds has been used once as a test sample. The corresponding training and test data are added to each of the molecule files, and the respective statistics data are collected at the bottom of Table 4. Again, due to the 10% smaller number of training molecules used in the 10 cross-validation calculations, the number of compounds for which the heat of combustion is evaluated as the test value is even smaller (lines E, F, G, and H) than that of the training set (lines B, C, and D). The statistics data of Table 4 also show a significantly lower number of “valid” groups in line A than the total number of atoms and special groups. The residual “invalid” groups, although at present not applicable for heat-of-combustion calculations, have been left in Table 4 for future use in this continuing project. Interested scientists may want to help to increase the number of “valid” groups in this database by molecules carrying the under-represented atom groups. At present, the list of elements for heat-of-combustion calculations is limited to H, B, C, N, O, P, S, Si, and/or halogen.

**Table 4 molecules-26-06101-t004:** Atom Groups and their Contributions to ΔH°(c) Calculations (in kJ/mol).

Entry	Atom Type	Neighbors	Contribution	Occurrences	Molecules
1	B	C3	−5771.41	10	10
2	B	C2O	−5234.3	2	2
3	B(-)	F4	−128.41	1	1
4	C sp3	H3B	927.34	3	1
5	C sp3	H3C	−774.53	5659	2598
6	C sp3	H3N	−1273.86	288	183
7	C sp3	H3N(+)	−1258.26	22	10
8	C sp3	H3O	−1273.9	493	333
9	C sp3	H3S	−1435.83	36	30
10	C sp3	H3P	−1106.35	3	1
11	C sp3	H3Si	−1323.2	116	49
12	C sp3	H2BC	1052.3	22	8
13	C sp3	H2C2	−653.47	9139	2066
14	C sp3	H2CN	−1150.33	632	346
15	C sp3	H2CN(+)	−1136.52	76	51
16	C sp3	H2CO	−1140.92	1209	753
17	C sp3	H2CS	−1313.58	180	118
18	C sp3	H2CP	−825.63	6	3
19	C sp3	H2CF	−626.92	15	14
20	C sp3	H2CCl	−616.8	81	70
21	C sp3	H2CBr	−620.8	23	20
22	C sp3	H2CJ	−685.85	12	9
23	C sp3	H2CSi	−1211.79	130	51
24	C sp3	H2N2	−1644.62	33	12
25	C sp3	H2N2(+)	−1666.85	6	6
26	C sp3	H2NO	−1630.65	8	6
27	C sp3	H2NS	−1776.97	2	1
28	C sp3	H2NS(+)	−1817.73	1	1
29	C sp3	H2NP(+)	−565.68	1	1
30	C sp3	H2O2	−1605.49	31	26
31	C sp3	H2OSi	−1715.41	1	1
32	C sp3	H2OCl	−1115.09	4	3
33	C sp3	H2S2	−1997.67	9	7
34	C sp3	HBC2	1197	6	2
35	C sp3	HC3	−529.09	1386	765
36	C sp3	HC2N	−1026.35	106	84
37	C sp3	HC2N(+)	−1004.99	43	40
38	C sp3	HC2O	−1010.37	545	330
39	C sp3	HC2S	−1179.91	34	25
40	C sp3	HC2Si	−1076.92	4	2
41	C sp3	HC2F	−486.29	5	5
42	C sp3	HC2Cl	−491.9	48	32
43	C sp3	HC2Br	−499.17	9	7
44	C sp3	HC2J	−574.72	4	3
45	C sp3	HCN2	−1514.29	5	4
46	C sp3	HCN2(+)	−1537.23	5	5
47	C sp3	HCNO	−1525.32	5	5
48	C sp3	HCNO(+)	−1522.69	4	2
49	C sp3	HCNS	−1683.66	4	2
50	C sp3	HCO2	−1473.47	63	54
51	C sp3	HCS2	−1791.66	1	1
52	C sp3	HCF2	−447.7	14	13
53	C sp3	HCFCl	−470.21	4	4
54	C sp3	HCCl2	−494.97	18	17
55	C sp3	HCClBr	−510.39	1	1
56	C sp3	HCBr2	−475.67	1	1
57	C sp3	HN3(+)	−2166.08	1	1
58	C sp3	HNO2	−1989.17	1	1
59	C sp3	HO3	−1920.79	6	6
60	C sp3	HOF2	−891.81	2	2
61	C sp3	BC3	1320.43	3	1
62	C sp3	C4	−403.7	392	287
63	C sp3	C3N	−886.44	46	34
64	C sp3	C3N(+)	−875.68	28	26
65	C sp3	C3O	−876.38	181	135
66	C sp3	C3S	−1050.27	23	19
67	C sp3	C3F	−451.56	11	6
68	C sp3	C3Cl	−355.26	9	9
69	C sp3	C3Br	−362.07	2	2
70	C sp3	C3J	−430.2	1	1
71	C sp3	C2N2(+)	−1417.05	9	9
72	C sp3	C2O2	−1331.4	42	38
73	C sp3	C2S2	−1708.54	4	1
74	C sp3	C2F2	−318.74	104	28
75	C sp3	C2FCl	−331.09	3	2
76	C sp3	C2Cl2	−357.37	7	7
77	C sp3	CN3(+)	−2020.09	19	11
78	C sp3	CN2F(+)	−1420.86	24	16
79	C sp3	CN2Cl(+)	−1451.19	2	2
80	C sp3	CNF2	−848.67	6	2
81	C sp3	CNF2(+)	−853.81	3	2
82	C sp3	CO3	−1771.57	8	7
83	C sp3	COF2	−802.95	3	3
84	C sp3	COCl2	−893.67	1	1
85	C sp3	CF3	−251.65	83	64
86	C sp3	CF2Cl	−306.09	10	8
87	C sp3	CF2Br	−319.62	5	4
88	C sp3	CFCl2	−317.2	7	7
89	C sp3	CFClBr	−276.51	1	1
90	C sp3	CCl3	−371.89	25	24
91	C sp3	CBr3	−345.19	1	1
92	C sp3	N2OF(+)	−1875.95	1	1
93	C sp3	N4(+)	−2635.7	3	3
94	C sp3	N3F(+)	−4981.42	2	2
95	C sp3	O4	−2239.99	3	3
96	C sp3	O2F2	−1255.75	1	1
97	C sp3	OF3	−692.57	2	2
98	C sp3	OF2Cl	−768.91	1	1
99	C(-) sp3	C3	−3078.32	2	2
100	C sp2	H2=C	−703.3	255	227
101	C sp2	H2=N	−1694.79	2	2
102	C sp2	HC=C	−563.48	1268	695
103	C sp2	HC=N	−1522.25	64	58
104	C sp2	HC=O	−390.29	115	111
105	C sp2	H=CN	−1024.99	141	103
106	C sp2	HC=N(+)	−5278.26	7	7
107	C sp2	H=CN(+)	−1032.17	4	4
108	C sp2	H=CO	−619.08	54	48
109	C sp2	H=CS	−1228.75	80	61
110	C sp2	H=CF	−547.73	2	2
111	C sp2	H=CCl	−550.31	8	6
112	C sp2	H=CBr	−574.22	2	2
113	C sp2	H=CSi	−1051.13	16	9
114	C sp2	HN=N	−1998.61	45	42
115	C sp2	HN=O	−830.28	12	11
116	C sp2	H=NO	−1583.32	2	2
117	C sp2	HO=O	−410.95	19	19
118	C sp2	H=NS	−2218.79	3	3
119	C sp2	C2=C	−430.98	318	255
120	C sp2	C2=N	−1378.53	82	67
121	C sp2	C2=N(+)	326.77	6	6
122	C sp2	C=CN	−893.41	86	66
123	C sp2	C=CN(+)	−928.07	10	10
124	C sp2	C2=O	−241.97	400	337
125	C sp2	C=CO	−470.12	86	69
126	C sp2	C=CS	−1085.43	56	45
127	C sp2	C=CF	−452.88	7	6
128	C sp2	C=CCl	−418.16	22	13
129	C sp2	C=CBr	−412.66	1	1
130	C sp2	=CN2	−1367.07	11	11
131	C sp2	=CN2(+)	−1387.08	10	10
132	C sp2	CN=N	−1858.11	48	40
133	C sp2	CN=N(+)	−1939.3	6	6
134	C sp2	CN=O	−687.04	310	243
135	C sp2	C=NO	−1412.25	18	16
136	C sp2	=CNO	−980.26	1	1
137	C sp2	=CNO(+)	−1004.7	6	6
138	C sp2	CN=S	−1516.78	7	6
139	C sp2	C=NS	−2037.97	6	6
140	C sp2	=CNS(+)	−1601.01	2	2
141	C sp2	=CNCl	−854.46	1	1
142	C sp2	CO=O	−256.51	1142	872
143	C sp2	CO=O(-)	98.04	51	50
144	C sp2	C=OS	−913.81	7	7
145	C sp2	C=OF	−193.64	3	3
146	C sp2	C=OCl	−202.04	14	11
147	C sp2	C=OBr	−203.56	2	2
148	C sp2	C=OJ	−281.05	2	2
149	C sp2	=COF	−297.83	2	2
150	C sp2	CS=S	−1716.14	3	3
151	C sp2	=CS2	−1853.93	2	1
152	C sp2	=CF2	−413.7	9	8
153	C sp2	=CFCl	−362.02	1	1
154	C sp2	=CCl2	−420.11	7	5
155	C sp2	=CJ2	−544.25	2	1
156	C sp2	N2=N	−2333.01	67	55
157	C sp2	N2=N(+)	583.11	2	2
158	C sp2	N2=O	−1148.85	124	107
159	C sp2	N=NO	−1909.16	3	3
160	C sp2	N2=S	−1999.02	27	25
161	C sp2	N=NS	−2485.54	10	9
162	C sp2	NO=O	−712.7	22	21
163	C sp2	N=OS	−1624.45	1	1
164	C sp2	NO=S	−1586.99	5	5
165	C sp2	=NOS	−2019.32	1	1
166	C sp2	=NOCl	−1416.86	1	1
167	C sp2	NS=S	−2180.9	6	6
168	C sp2	NS=S(-)	−2015.49	4	4
169	C sp2	=NSCl	−2036.68	1	1
170	C sp2	O2=O	−288.27	14	14
171	C sp2	O=OCl	−207.85	4	4
172	C sp2	=OS2	−1589.52	2	2
173	C sp2	S2=S	−2384.68	3	3
174	C aromatic	H:C2	−544	10,741	1946
175	C aromatic	H:C:N	−677.57	176	121
176	C aromatic	H:C:N(+)	−664.15	46	25
177	C aromatic	H:N2	−805.13	12	10
178	C aromatic	:C3	−404.91	496	193
179	C aromatic	C:C2	−412.39	2572	1349
180	C aromatic	C:C:N	−537.55	106	62
181	C aromatic	C:C:N(+)	−537.4	37	21
182	C aromatic	:C2N	−904.12	521	380
183	C aromatic	:C2N(+)	−924.19	323	214
184	C aromatic	:C2:N	−541.18	73	54
185	C aromatic	:C2:N(+)	−537.03	33	18
186	C aromatic	:C2O	−485.35	724	496
187	C aromatic	:C2P	−739.79	9	3
188	C aromatic	:C2S	−1093.91	94	75
189	C aromatic	:C2Si	−977.52	30	11
190	C aromatic	:C2F	−400.21	136	67
191	C aromatic	:C2Cl	−391.36	235	137
192	C aromatic	:C2Br	−393.11	72	50
193	C aromatic	:C2J	−466.27	39	34
194	C aromatic	C:N2	−653.04	5	3
195	C aromatic	:CN:N	−1014.82	17	13
196	C aromatic	:CN:N(+)	−1105.83	3	2
197	C aromatic	:C:NO	−567.84	11	11
198	C aromatic	:C:NCl	−521.23	30	21
199	C aromatic	:C:NBr	−517.98	4	3
200	C aromatic	N:N2	−1126.33	22	14
201	C aromatic	:N2O	−708.43	17	6
202	C aromatic	:N2S	−1372.64	1	1
203	C aromatic	:N2Cl	−639.79	11	10
204	C(+) aromatic	H:N2	915.78	17	17
205	C(+) aromatic	:N3	1847.84	3	3
206	C sp	H#C	−654.9	50	42
207	C sp	C#C	−502.89	198	108
208	C sp	=C2	−532.17	12	11
209	C sp	C#N	−495.27	165	139
210	C sp	C#N(+)	−521.62	4	3
211	C sp	C#N(-)	378.06	6	2
212	C sp	#CN	−1069.64	2	2
213	C sp	=C=N	−1519.98	2	2
214	C sp	=C=O	−281.24	4	3
215	C sp	#CS	−1214.94	2	2
216	C sp	#CCl	−514.93	3	2
217	C sp	#CSi	−1091.51	3	3
218	C sp	N#N	−982.36	4	4
219	C sp	N#N(-)	−144.09	10	5
220	C sp	=N2	−2404.6	2	2
221	C sp	#NO	−648.9	2	2
222	C sp	=N=O	−1216.26	22	16
223	C sp	#NS	−1277.41	1	1
224	C sp	=N=S	−2056.03	2	2
225	C sp	=N=S(-)	−1076.3	2	2
226	N sp3	H2C	218.81	64	56
227	N sp3	H2C(pi)	253.54	334	285
228	N sp3	H2N	−304.07	29	23
229	N sp3	H2N(pi)	−266.71	1	1
230	N sp3	H2S	215.36	9	9
231	N sp3	HC2	814.55	69	63
232	N sp3	HC2(pi)	846.53	138	105
233	N sp3	HC2(2pi)	845.11	253	200
234	N sp3	HCN	288.21	5	3
235	N sp3	HCN(pi)	315.32	41	28
236	N sp3	HCN(+)(pi)	734.53	5	4
237	N sp3	HCN(2pi)	359.34	69	64
238	N sp3	HCN(+)(2pi)	717.66	6	6
239	N sp3	HCO(pi)	520.5	2	2
240	N sp3	HCS(pi)	1015.17	3	3
241	N sp3	HCSi	829.19	5	5
242	N sp3	HN2(2pi)	−176.66	1	1
243	N sp3	HNS	552.34	1	1
244	N sp3	HSi2	850.75	1	1
245	N sp3	C3	1409.08	84	73
246	N sp3	C3(pi)	1429.38	98	84
247	N sp3	C3(2pi)	1430.98	69	52
248	N sp3	C3(3pi)	1421.7	31	23
249	N sp3	C2N	871.27	1	1
250	N sp3	C2N(pi)	896.78	13	11
251	N sp3	C2N(+)(pi)	1320.4	40	25
252	N sp3	C2N(2pi)	954	23	22
253	N sp3	C2N(+)(2pi)	1269.38	12	7
254	N sp3	C2N(3pi)	948.11	9	9
255	N sp3	C2N(+)(3pi)	1230.43	3	3
256	N sp3	C2O	1037.7	3	3
257	N sp3	C2S	584.44	6	3
258	N sp3	C2Si	1437.12	8	6
259	N sp3	C2F(2pi)	−2337.09	1	1
260	N sp3	C2Cl(2pi)	878.7	1	1
261	N sp3	C2Br(2pi)	900.67	1	1
262	N sp3	CN2(2pi)	491.47	9	7
263	N sp3	CN2(+)(2pi)	1183.51	1	1
264	N sp3	CN2(3pi)	550.35	3	3
265	N sp3	CN2(+)(3pi)	774.31	3	3
266	N sp3	CF2	197.51	12	7
267	N sp3	CF2(pi)	997.31	1	1
268	N sp3	Si3	1479.07	1	1
269	N sp2	H=C	760	10	10
270	N sp2	C=C	1411.35	154	133
271	N sp2	C=N	375.86	70	38
272	N sp2	C=N(+)	714.59	35	31
273	N sp2	=CN	866.59	141	117
274	N sp2	=CN(+)	1299.9	5	5
275	N sp2	C=O	421.69	13	12
276	N sp2	=CO	935.48	78	55
277	N sp2	=CS	705.75	2	1
278	N sp2	=CF	0	1	1
279	N sp2	N=N	−82.18	80	41
280	N sp2	N=O	1.35	8	6
281	N sp2	=NO	762.47	2	1
282	N sp2	=NO(+)	1041.44	11	6
283	N sp2	O=O	831.52	9	9
284	N sp2	P=P	−482.14	7	2
285	N aromatic	H2:C(+)	−1025.73	5	3
286	N aromatic	HC:C(+)	−363.24	2	2
287	N aromatic	C2:C(+)	216.63	36	19
288	N aromatic	:C2	214.34	273	189
289	N aromatic	:C:N	41.42	6	3
290	N aromatic	:C:N(+)	2190.8	1	1
291	N(+) sp3	H3C	57.66	47	46
292	N(+) sp3	H2C2	607.96	9	9
293	N(+) sp3	HC3	1364.55	6	4
294	N(+) sp3	C4	1885.72	8	8
295	N(+) sp2	C=CO(-)	5214.75	7	7
296	N(+) sp2	C=NO	442.92	16	8
297	N(+) sp2	C=NO(-)	155.64	16	11
298	N(+) sp2	CO=O(-)	548.28	550	310
299	N(+) sp2	=CO2(-)	−568.22	6	6
300	N(+) sp2	NO=O(-)	−366.72	76	54
301	N(+) sp2	O2=O(-)	188.79	73	37
302	N(+) aromatic	C:C2	698.75	1	1
303	N(+) aromatic	:C2O(-)	234.67	58	40
304	N(+) aromatic	:C:NO(-)	−2193.1	1	1
305	N(+) sp	C#C(-)	−94.07	6	6
306	N(+) sp	#CO(-)	0	4	3
307	N(+) sp	=N2(-)	−542.94	30	26
308	N(-)	C2	−776.85	5	5
309	O	HC	550.37	663	373
310	O	HC(pi)	149.9	795	622
311	O	HN	−183.46	3	3
312	O	HN(pi)	−66.29	29	23
313	O	HO	−35.98	29	26
314	O	HP	−107.27	3	2
315	O	HS	346.8	8	8
316	O	HSi	241.58	1	1
317	O	BC	1904.24	2	2
318	O	C2	1101.66	471	283
319	O	C2(pi)	701.99	896	686
320	O	C2(2pi)	278.25	167	156
321	O	CN(pi)	−291.5	24	18
322	O	CN(+)(pi)	401.59	63	29
323	O	CN(2pi)	131.93	14	14
324	O	CN(+)(2pi)	398.5	1	1
325	O	CO	523.43	120	65
326	O	CO(pi)	113.28	65	29
327	O	CS	457.66	18	9
328	O	CP	542.76	10	4
329	O	CP(pi)	91.26	3	1
330	O	CSi	708.96	54	21
331	O	CSi(pi)	318.69	38	15
332	O	N2(2pi)	−65.35	15	14
333	O	N2(+)(2pi)	−220.1	5	5
334	O	OSi	106.92	8	4
335	O	Si2	400.13	11	3
336	P3	C3	124.54	3	3
337	P4	C3=O	−243.18	1	1
338	P4	C3=S	−373.61	1	1
339	P4	C2O=O	−169.24	1	1
340	P4	CO2=O	197.04	1	1
341	P4	CO2=O(-)	−394.07	1	1
342	P4	N=NCl2	0	7	2
343	P4	O3=O	14.1	4	4
344	S2	HC	−88.46	47	42
345	S2	HC(pi)	−58.54	10	10
346	S2	C2	690.55	78	66
347	S2	C2(pi)	714.51	26	21
348	S2	C2(2pi)	750.83	88	82
349	S2	CN(pi)	−618.03	1	1
350	S2	CS	42.29	18	9
351	S2	CS(pi)	53.77	16	8
352	S2	N2	25.26	1	1
353	S2	N2(2pi)	0	1	1
354	S2	NS	−291.87	2	1
355	S4	C2=O	849.53	8	8
356	S4	C2=O2	1073.53	43	43
357	S4	CN=O2	−41.35	11	11
358	S4	CO=O2	216.14	3	3
359	S4	CO=O2(-)	777.27	2	2
360	S4	C=O2S	394.52	2	1
361	S4	N2=O2	558.49	1	1
362	S4	NO=O2	−918.69	1	1
363	S4	O2=O	−92.99	5	5
364	S4	O2=O2	116.23	4	4
365	S4	O2=O2(-)	−556.38	4	4
366	S4	O=O2F	−470.05	1	1
367	S4	O=O2Cl	−463.63	1	1
368	Si	H3C	−740.19	4	4
369	Si	H2C2	12.42	2	2
370	Si	HC3	602.13	29	29
371	Si	HC2Cl	67.18	1	1
372	Si	HCCl2	−100.05	1	1
373	Si	HN3	−2430.7	1	1
374	Si	HO3	−931.42	1	1
375	Si	C4	1327.16	15	15
376	Si	C3N	317.05	15	12
377	Si	C3O	813.97	12	12
378	Si	C3Cl	1013.39	1	1
379	Si	C3Br	1000.85	1	1
380	Si	C2O2	285.19	16	8
381	Si	C2Cl2	592.69	3	3
382	Si	CO3	−235.18	16	16
383	Si	CCl3	145.43	1	1
384	Si	O4	−763.98	7	7
385	H	H Acceptor	0.27	241	188
386	H	.H	−5.79	381	142
387	H	..H	−1.31	4908	1297
388	Angle60		−35.25	405	118
389	Angle90		−24.51	321	66
390	Angle102		−4.65	1663	451
A	Based on	Valid groups	267		5030
B	Goodness of fit	R^2^	1		4886
C	Deviation	Average	13.66		4886
D	Deviation	Standard	18.12		4886
E	K-fold cv	K	10		4790
F	Goodness of fit	Q^2^	1		4790
G	Deviation	Average (cv)	14.44		4790
H	Deviation	Standard (cv)	19.16		4790

## 3. Sources of Heat-of-Combustion and Formation Data

The present list of references encompasses the sources for the experimental standard heats of combustion as well as those of formation, because the input of the heat of combustion into a molecule’s database immediately also triggers the calculation and addition of its heat of formation and vice versa. Experimental data given in kcal/mol are translated into kJ/mol by multiplication with 4.1858.

A large number of experimental data have been provided by several comprehensive papers; in particular, Domalski’s collection [21] published an extended variation of compounds containing the elements C, H, N, O, P, and S. The CRC *Handbook of Chemistry and Physics* [22] included a chapter containing the heats of formation of another large list of compounds. In the last 6 years since the publication of the predecessor version [1] of this paper, a large number of publications have been found, which produced further experimental combustion and formation data. In the following, they have been sorted by their dominant contributory structural features to the present subject. An especially extended amount of research has been done with hydrocarbons including alkanes, alkenes, alkynes, and aromatics [23,24,25,26,27,28,29,30,31,32,33,34,35,36,37,38,39,40,41,42,43,44,45,46,47,48,49,50,51,52,53,54,55,56,57,58,59,60], forming the core of the various carbon groups. In addition, many data have dealt specifically with alcohols and phenol derivatives [61,62,63,64,65,66,67,68,69,70,71,72,73,74], ethers [75,76,77,78,79,80,81,82,83,84], carbaldehydes [85,86,87,88,89,90,91,92,93,94], ketones [95,96,97,98,99,100,101,102,103,104,105,106,107,108,109,110,111,112], carboxylic acids [113,114,115,116,117,118,119,120,121,122,123,124,125,126,127,128,129,130,131,132,133,134,135], carboxylic esters, carbonates and lactones [136,137,138,139,140,141,142,143,144,145,146,147,148,149,150,151,152,153,154], sugars [155], peroxides [156,157,158,159,160,161], amines and imines [162,163,164,165,166,167,168,169], amides, imides, amidines and hydrazides [170,171,172,173,174,175,176,177,178,179,180,181,182,183,184,185,186], guanidines [187,188], ureas [189,190,191,192,193,194,195,196], urethanes [197], carbamates [198], azides [199,200], nitriles and nitriloxides [201,202,203,204,205,206,207,208], isocyanates [209], oximes [210], nitramines [211], azo- and azoxy compounds [212,213,214], *N*-oxides [215,216,217,218,219,220,221,222,223,224,225,226,227,228,229,230], nitroso [231] and nitro compounds [232,233,234,235,236,237,238,239,240,241,242,243,244,245,246,247,248,249,250,251,252,253,254,255,256,257,258,259,260,261,262,263], nitrates [264], amino acids [265,266,267,268,269,270,271,272,273,274,275,276,277], sulfur-containing [278,279,280,281,282,283,284,285,286,287,288,289], phosphorus-containing [290], silicon-containing [291,292,293,294], and boron-containing compounds [295]. Beyond these, a large number of halogen-substituted compounds, many of them carrying any of the further functional groups just mentioned, have been found [296,297,298,299,300,301,302,303,304,305,306,307,308,309,310,311,312,313,314,315,316,317,318,319,320,321,322,323,324,325,326,327,328,329,330,331,332,333,334,335,336,337,338,339,340,341,342,343,344,345,346,347,348,349,350,351,352,353,354,355,356,357]. A considerable number of experimental combustion and formation data have been published for heterocyclic compounds, including hetarenes, unsubstituted and substituted by functional groups just mentioned. According to the hetero elements in the ring system, they have been subdivided into N_x_-heterocycles (where x is 1 to 4) [358,359,360,361,362,363,364,365,366,367,368,369,370,371,372,373,374,375,376,377,378,379,380,381,382,383,384,385,386,387,388,389,390,391,392,393,394,395,396,397,398,399,400,401,402,403,404,405,406,407,408,409,410,411,412,413,414,415,416,417,418,419,420,421,422,423], N,O-heterocycles [424,425,426,427,428,429,430,431,432,433], N,S-heterocycles [434,435,436,437,438], O_x_-heterocycles [439,440,441,442,443,444,445,446,447,448,449,450,451,452], and S_x_-heterocycles [280,282,283,284,453,454,455,456,457,458,459,460,461]. A small number of papers contributed data for hetarenes with several element combinations [462,463,464,465,466,467,468,469,470,471]. In addition, and as an important extension to the earlier paper [1], a great variety of ionic liquids has been added [472,473,474,475,476,477,478,479,480,481,482,483,484,485,486,487,488,489,490,491,492,493,494,495,496,497,498]. Finally, a number of publications contributed combustion and/or formation data that could not be assigned to any of the aforementioned classes [499,500,501,502,503,504,505,506,507,508,509,510,511,512,513,514,515,516,517,518,519,520,521,522,523,524,525,526].

## 4. Results

### 4.1. Heat of Combustion

The first preliminary calculations of the group contributions were based on the complete set of 5560 compounds for which experimental heats of combustion and/or formation were available. However, contrary to the approach in the earlier paper [1], a further restriction was introduced in that only those compounds were allowed to remain in the consecutive calculations, the experimental values of which did not deviate by more than three times the cross-validated standard error from the cross-validated calculated value. Accordingly, the final group contributions rested on 5030 compounds, as shown on row A in Table 4. The discarded molecules have been collected in an outliers list, available with Appendix A. As a consequence, the correlation coefficient Q^2^ is even better than the previously published value of 0.9999 and is now indistinguishable from 1 (row F in Table 4). Analogously, the new cross-validated standard error of 19.16 kJ/mol (row H in Table 4) is considerably better than the earlier one of 25.2 kJ/mol. Not surprisingly, the mean absolute deviation over 4886 compounds is just 0.4% over a calculated heat-of-combustion range of from −72 kJ/mol (hydrogen peroxide) to −35,112.2 kJ/mol (glycerol trioleate). These excellent statistical data are well reflected in the straight line of the data points in the correlation diagram of Figure 1 and the perfectly symmmetrically balanced Gaussian bell curve of the histogram in Figure 2. The only downside, however, is the much longer list of 390 atoms and special groups required (compared to the 273 of the earlier paper [1]), of which only 267 are “valid” for predictions. However, the latter still enable the calculation of the heats of combustion and formation of presently 29,067 molecules, i.e., ca. 84.5% of the complete dataset. The complete set of molecules used for the group-parameters calculations is available in the Appendix A.

The extraordinary accuracy of the predictions allows a deeper analysis of the actual structural state of certain classes of molecules for which alternative structures are possible at standard conditions, in particular as to which prototropic forms are prevailing in amino acids and which tautomeric form is prevalent in compounds that may exist in both hydroxyazo and hydrazone or keto and enol forms. Beyond this, an educated estimate as to what the enthalpy difference is between the alternative forms might be possible.

#### 4.1.1. Amino Acids

It is common knowledge that amino acids exist in zwitterionic form both in the crystalline as well as the liquid state [527], whereas in the gas phase they exist in their non-ionic form. To our knowledge, the difference in the enthalpies of combustion between these two forms has not yet been systematically analyzed. In Table 5, the calculated values for the non-ionic and zwitter-ionic forms of a series of amino acids are compared with their experimental data.

The average ΔH°(c) difference was calculated as ca. 61.5 kJ/mol, with the non-ionic form exhibiting the more negative value. Cystine is an outlier in that it contains two amino-acid functions. Interestingly, sarcosine (*N*-methylglycine) shows the lowest difference between the two forms, which is due to the fact that it carries a less basic dialkylamino group. Similarly, *N*-phenylglycine differs from the remaining amino acids by an amino group that is conjugated to the phenyl ring, again lowering its basicity. Except for these special cases, the experimental values are in better compliance with the calculated values of the zwitter forms.

#### 4.1.2. Azo-Hydrazone Tautomerism

The observation of the hydroxyazo-hydrazone tautomerism is well known among dye chemists dealing with azo dyes, as it has a drastic effect on the electronic absorption spectra. In an earlier paper [1], it was demonstrated that the direction of the tautomeric equilibrium is fairly predictable on the basis of the calculated heats of formation of the hydroxyazo and the hydrazone form. Analogously, the heats of combustion, now founded on a much larger structural basis, should confirm these observations, with the less negative enthalpy indicating the dominating form. Indeed, in conformance with experimental observation, the calculated values listed in Table 6 confirm that arylazo-naphthols primarily exist in their hydrazone form, whereas the opposite is true for the arylazo-naphthylamines. On the other hand, the small enthalpy difference found between the two forms of the phenylazophenols confirms their weak tendency to tautomerize. In addition, the available experimental heats of combustion for 4-phenylazophenol and 4-aminoazobenzene are in fairly good agreement with their prevailing forms.

#### 4.1.3. Keto-Enol Tautomerism

Prediction of the dominant forms in keto-enol tautomers under standard conditions has been shown to be at best coincidental in [1], which is not surprising in view of the mostly small enthalpy differences between the two forms. Recalculated values of the heat of combustion of the example molecules in [1], based on the updated group-parameters set, are compared with their experimental values, where available, in Table 7. As is evident, except for acetone, the enol form is supposed to be the dominant tautomer throughout, which clearly contradicts the experience, most prominently with cyclohexanone and cyclopentanone. Beyond this, the experimental values are of no help despite the small standard error Q^2^ of 19.16 kJ/mol (see Table 4) because the deviations between the enthalpies of both forms with the experiment are well within the tolerated boundaries.

#### 4.1.4. Ionic Liquids

The main extension of the present atom-groups additivity method enabled the inclusion of the heats of combustion of the ionic liquids. Unfortunately, of the 679 ionic liquids presently stored in the database, only for 28 of them was the experimental heat of combustion comparable with calculated values to this date due to the restrictions mentioned earlier. They essentially cover nitrates, dicyanamides, sulfates, dialkyldithiocarbamates, and halogenides of various imidazolium, ammonium, and glycinium cations. In Table 8, these compounds are listed, and their experimental values are compared with the calculated ones. Their conformance is exceptionally good, resulting in a mean absolute deviation of only 0.23%.

### 4.2. Heat of Formation

The heat of formation has been calculated indirectly from the calculated heat of combustion for each compound for which experimental data were available using the heats of combustion for the elements given in [20,21]. Accordingly, the same restrictions concerning “te” valid “ty” of the atom groups as well as the elements themselves apply. Therefore, the number of compounds in the correlation diagram of Figure 3 is identical with that of Figure 1. However, due to the distinctly smaller range of heat-of-formation values from −7238.2 (perfluorohexadecane) to +1039.7 kJ/mol (2,4,6-triazido-s-triazine) and the error-propagation effect, the correlation coefficient R^2^ is “only” 0.9979, and since the standard error σ is still 18.14 kJ/mol, their mean absolute deviation is 27.23%. The histogram of Figure 4 again confirms the symmetrical Gaussian error distribution of the experimental heats of formation about the calculated ones.

## 5. Conclusions

The present paper is proof of the easy expandability of the group-additivity method outlined in [1] for the calculation of the heats of combustion and formation of in principle any organic molecule to consider. A large amount of more than 5000 molecules upon which the atom-group parameters are based allowed strict filtering out of the worst outliers without undue sacrifice of “invalidated” atom groups, resulting in an as-yet unsurpassed accuracy of the predicted heat of combustion with a mean absolute deviation of only 0.4% for up to 84.5% of nearly any kind of organic compound. Beyond this, the present method basically allows the accurate calculation of a molecule’s heat of combustion simply by means of paper and pencil, using the presented group parameters in Table 4. As this work is ongoing, the number of compounds for which—based on the same algorithm—up to 17 physical, thermodynamic, solubility-, optics-, charge-, and environment-related descriptors [1,2,3,4,5,6] can be reliably predicted, will steadily increase.

The mentioned software project is called ChemBrain IXL, available from Neuronix Software (www.neuronix.ch, Rudolf Naef, Lupsingen, Switzerland).

## Figures and Tables

**Figure 1 molecules-26-06101-f001:**
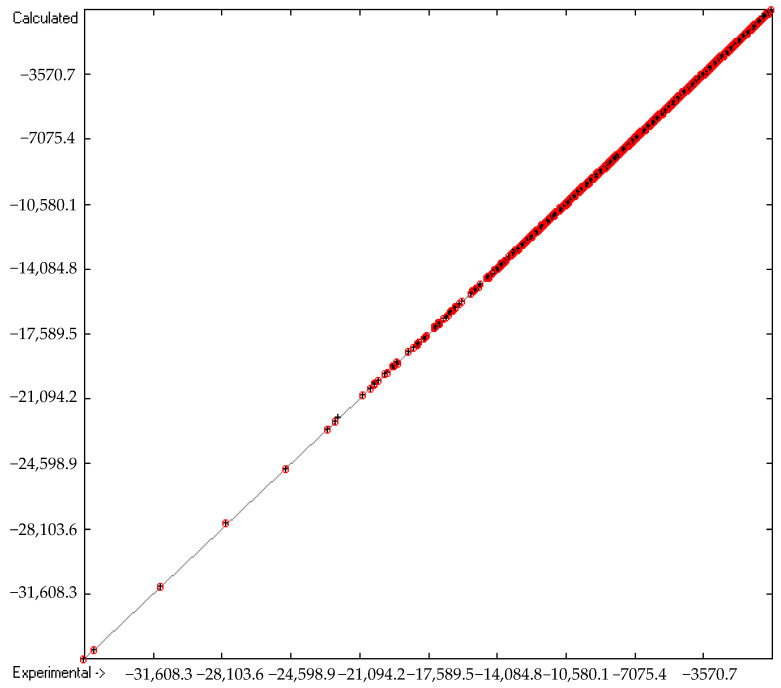
Correlation diagram of the heat-of-combustion data in kJ/mol. Cross-validation data are added as red circles. (10-fold cross-validated: N = 4886, Q^2^ = 1, regression line: intercept = −1.0111; slope = 0.9998).

**Figure 2 molecules-26-06101-f002:**
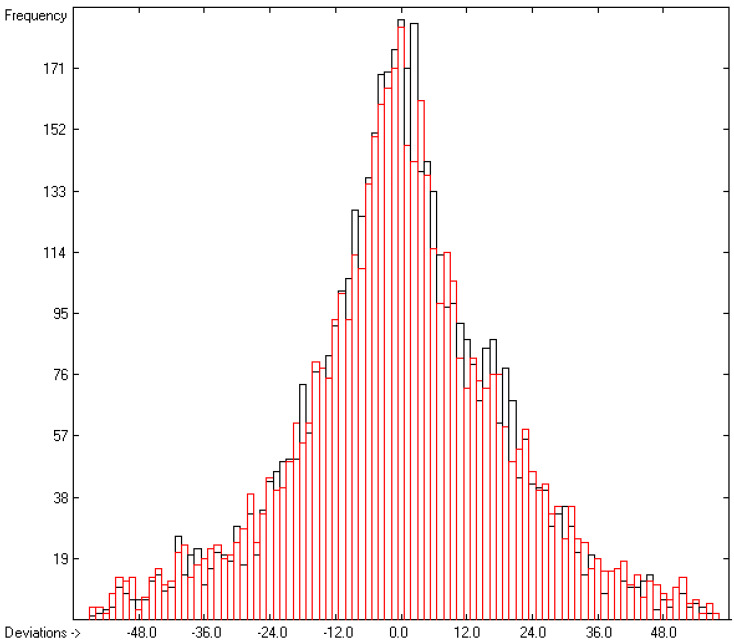
Histogram of the heat-of-combustion data in kJ/mol. Cross-validation data are superpositioned as red bars. (σ = 18.12; S = 19.16; experimental values range from −35,100 to −98.2).

**Figure 3 molecules-26-06101-f003:**
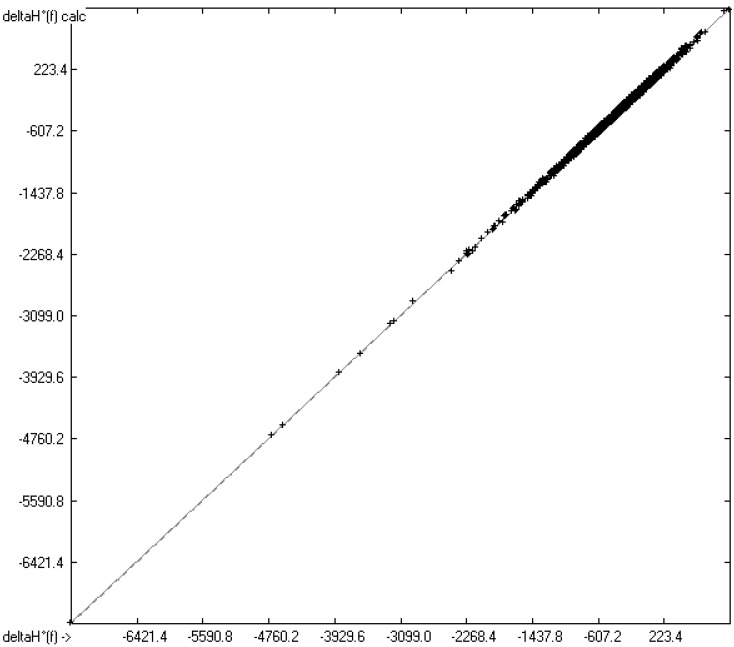
Correlation diagram of the heat of formation (in kJ/mol), (N = 4886, R^2^ = 0.9979, regression line: intercept = −0.5539; slope = 0.9979).

**Figure 4 molecules-26-06101-f004:**
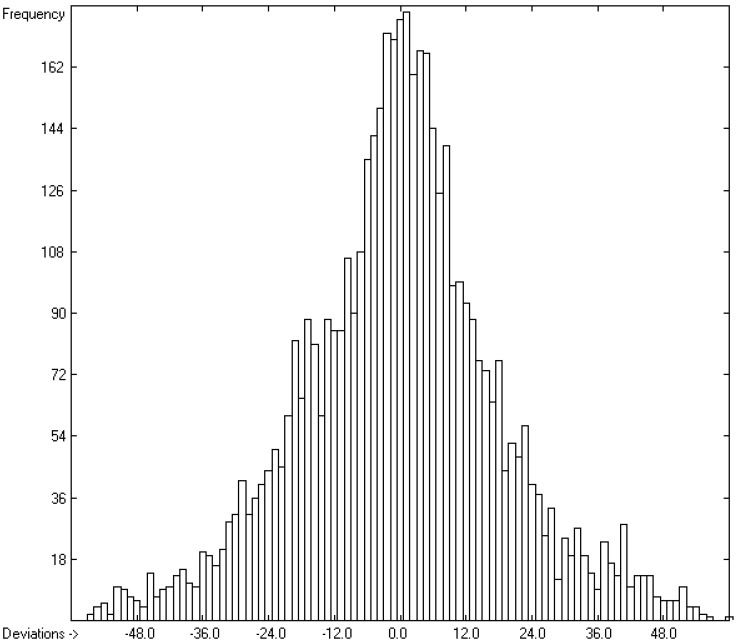
Histogram of the heat of formation (in kJ/mol) (σ = 18.14; experimental values range from −7238.2 to +1039.7 kJ/mol).

**Table 1 molecules-26-06101-t001:** Charged Atom Groups and their Meaning.

No	Atom Type	Neighbors	Meaning	Example
1	B(-)	F4	BF_4_^-^	tetrafluoroborate
2	C(-) sp3	C3	C-C^-^(C)-C	tricyanomethanide
3	C sp2	NS=S(-)	N-C(=S)-S^-^	dithiocarbamate
4	C aromatic	H:C:N(+)	C:CH:N^+^	C2 in pyridinium
5	C(+) aromatic	H:N2	N:C^+^(H):N	C2 in imidazolium
6	C sp	C#N(-)	N#C-C^-^	tricyanomethanide
7	C sp	N#N(-)	N#C-N^-^	dicyanoamide
8	C sp	=N=S(-)	N=C=S^-^	Thiocyanate
9	N(+) sp3	C4	NC_4_^+^	tetraalkylammonium
10	N(+) sp2	O2=O(-)	NO_3_^-^	nitrate
11	N aromatic	C2:C(+)	(C)(C):C^+^	N1 in 1-alkylimidazolium
12	N(+) aromatic	C:C(C):N^+^(C):C	N in 1-alkylpyridinium	
13	N(-)	C2	C-N^-^-C	dicyanoamide
14	S4	O2=O2(-)	SO_4_^-^	hydrosulfate
15	S4	CO=O2(-)	C-SO3^-^	methylsulfonate

**Table 2 molecules-26-06101-t002:** Special Groups and their Meaning.

Atom Type	Neighbors	Meaning
H	H Acceptor	Intramolecular H bridge between acidic H (on O, N or S) and basic acceptor (O, N or F) at distance <1.75 Angstroms
H	.H	Intramolecular H–H distance <2 Angstroms
H	..H	Intramolecular H–H distance 2–2.3 Angstroms
Angle60		Bond angle <74 deg
Angle90		Bond angle 74–98 deg
Angle102		Bond angle 98–106 deg

**Table 3 molecules-26-06101-t003:** Example Calculation of the Standard Heat of Combustion (in kJ/mol) of 4-Methylene-2-oxetanone.

Atom Type Neighbors	C sp3 H2C2	C sp2 H2=C	C sp2 C=CO	C sp2 CO=O	O C2(2pi)	Angle90	Sum
Contribution	−653.47	−703.3	−470.12	−256.51	278.25	−24.51	
n Groups	1	1	1	1	1	4	
N × Contrib.	−653.47	−703.3	−470.12	−256.51	278.25	−98.04	−1903.19

**Table 5 molecules-26-06101-t005:** Calculated ΔH°(c) (in kJ/mol) of Non-ionic and Zwitter-ionic Forms of Amino Acids.

Molecule name	ΔH°_c_ calc Non-Ionic Form	Diff.	ΔH°_c_ calc Zwitter Form	ΔH°_c_ exp	References
(l)-Alanine	−1688.7	−64.9	−1623.8	−1621.0	[267,268]
(l)-Cy(l)-Cysteine	−2316.2	−64.9	−2251.3	−2263.0	[21]
(l)-Cystine	−4373.5	−132.3	−4241.2	−4248.0	[21]
(l)-Histidine	−3230.8	−64.9	−3165.9	−3180.6	[275]
(l)-Hydroxyproline	−2605.5	−37.9	−2567.6	−2594.1	[21]
(l)-Methionine	−3626.5	−62.3	−3564.2	−3564.1	[266]
2-Aminobutyric acid	−2342.1	−62.2	−2279.9	−2254.0	[21]
2-Methylalanine	−2323.3	−54.3	−2269.0	−2265.9	[21]
2-Phenylglycine	−4046.5	−62.2	−3984.3	−4005.1	[21]
4-Aminobutyric acid	−2345.1	−57.3	−2287.8	−2283.9	[21]
5-Aminovaleric acid	−2998.5	−57.3	−2941.2	−2937.0	[21]
8-Aminocaprylic acid	−4958.9	−57.3	−4901.6	−4884.0	[21]
Asparagine	−2000.8	−64.8	−1936.0	−1928.5	[265,266,269]
Aspartic acid	−1674.0	−64.9	−1609.1	−1602.9	[21]
beta-Alanine	−1691.6	−57.3	−1634.3	−1622.9	[21]
Dopa	−4285.5	−67.5	−4218.0	−4177.8	[21]
epsilon-Aminocaproic acid	−3652.0	−57.3	−3594.7	−3582.2	[21]
Glutamic acid	−2327.7	−62.2	−2265.5	−2277.0	[21]
Glutamine	−2654.6	−62.3	−2592.3	−2572.8	[265,269]
Glycine	−1038.1	−57.3	−980.8	−978.6	[268,269]
Isoleucine	−3651.0	−62.2	−3588.8	−3583.7	[269]
Isoserine	−1497.9	−57.4	−1440.5	−1438.2	[21]
Leucine	−3648.4	−64.9	−3583.5	−3581.2	[269]
Norleucine	−3649.1	−62.3	−3586.8	−3582.2	[21]
*N*-Phenylglycine	−4074.7	−40.2	−4034.5	−4037.6	[21]
Phenylalanine	−4702.6	−67.5	−4635.1	−4646.3	[269]
Proline	−2798.9	−56.5	−2742.4	−2746.2	[269]
Sarcosine	−1716.2	−27.4	−1688.8	−1675.1	[270]
Serine	−1504.4	−64.8	−1439.6	−1438.9	[269]
Threonine	−2151.3	−62.2	−2089.1	−2087.1	[269,275,277]
Tryptophane	−5671.0	−64.9	−5606.1	−5629.4	[269,274]
Tyrosine	−4494.1	−64.9	−4429.2	−4428.1	[269]
Valine	−2994.9	−62.2	−2932.7	−2933.9	[269]

**Table 6 molecules-26-06101-t006:** Calculated ΔH°(c) (in kJ/mol) of Azo and Hydrazone Forms of some Azo Dyes.

Compound	Hydrazone Form ∆H_c_ calc	Azo Form ∆H_c_ calc	∆H_c_ exp	^a^	Ref.
4-Phenylazophenol	−6275.2	−6288.0	−6314.1	+ −	[528]
2-Phenylazophenol	−6272.3	−6287.2	-	+ −	[528]
4-Aminoazobenzene	−6651.1	−6603.1	−6617.4	+	[529]
2-Aminoazobenzene	−6648.4	−6602.4		+	
1-Phenylazo-2-naphthol	−8145.8	−8185.1		+	[530,531]
4-Phenylazo-1-naphthol	−8148.5	−8185.4		+	[532]
1-Phenylazo-2-naphthylamine	−8533.8	−8500.3		+	[530,531]
4-Phenylazo-1-naphthylamine	−8524.6	−8503.0		+	[533]

^a^ Conformance with experimental data.

**Table 7 molecules-26-06101-t007:** Calculated and experimental ΔH°(c) (in kJ/mol) of Tautomeric Ketones and β-Diketones.

Compound	Keto Form ∆H_c_ ealc	Enol Form ∆H_c_ ealc	∆H_c_ exp	^a^	Ref.
1-(*N*-Phenylformimidoyl)-2-naphthol	−8608.3	−8560.3		+	[534]
Acetone	−1791.0	−1798.0	−1816.5	+	[535]
Cyclohexanone	−3509.3	−3497.6	−3517.6	−	[535]
Cyclopentanone	−2865.1	−2858.1	−2873.5	−	[536]
Phenol	−3149.4	−3055.4	−3055.5	+	[537]
2-Pyridone	−2557.4	−2513.2	−2517.62	−	[538,539,540]
4-Pyridone	−2573.8	−2564.2	−2537.5	+	[538,539,540]
Carbostyril	−4461.4	−4413.7	−4397.1	−	[541,542,543]
Acetylacetone	−2686.5	−2674.5	−2687.0	+	[544]
Bis(trifluoroacetyl)methane	−1640.7	−1628.7	−1673.7	+	[544]
Dibenzoylmethane	−7404.8	−7394.1	−7398.5	+	[544]
1,1-Bis(benzoyl)ethane	−8057.6	−8036.1		−	[544]

^a^ Conformance with experimental data.

**Table 8 molecules-26-06101-t008:** Calculated and experimental ΔH°(c) (in kJ/mol) of some Ionic Liquids.

Molecule Name	ΔH°_c_ exp	ΔH°_c_ calc	Deviation	Dev. in %
1,1,3,3-Tetramethylguanidinium nitrate	−3656.5	−3656.5	0.0	0.00
1-Butyl-1-methylpyrrolidinium dicyanamide	−7244.8	−7250.1	5.3	−0.07
1-Butyl-3-methylimidazolium chloride	−5232.3	−5206.6	−25.7	0.49
1-Butyl-3-methylimidazolium dicyanoamide	−6273.9	−6271.6	−2.3	0.04
1-Butyl-3-methylimidazolium nitrate	−5013.2	−5017.8	4.6	−0.09
1-Decyl-3-methylimidazolium bromide	−9105.2	−9127.4	22.2	−0.24
1-Dodecyl-3-methylimidazolium bromide	−10,406.0	−10,434.4	28.4	−0.27
1-Ethanol-3-methyl-imidazolium dicyanoamide	−4793.0	−4780.7	−12.3	0.26
1-Ethyl-3-methylimidazolium chloride	−3886.2	−3899.7	13.5	−0.35
1-Ethyl-3-methylimidazolium dicyanamide	−4955.4	−4964.7	9.3	−0.19
1-Ethyl-3-methylimidazolium nitrate	−3697.5	−3710.9	13.4	−0.36
1-Methyl-3-pentylimidazolium chloride	−5904.3	−5860.1	−44.2	0.75
1-Octyl-3-methylimidazolium bromide	−7837.8	−7820.5	−17.3	0.22
1-Tetradecyl-3-methylimidazolium bromide	−11,718.0	−11,741.3	23.3	−0.20
6,6-(Tetramethylene-3′-oxa)-7a-(nitroxymethyl)-3-oxoperhydroimidazo [1,5-c]oxazol-6-ium nitrate	−5384.8	−5376.5	−8.3	0.15
6,6-(Tetramethylene-3′-oxa)-7a-methyl-3-oxoperhydroimidazo [1,5-c]oxazol-6-ium nitrate	−5587.5	−5604.8	17.3	−0.31
6,6-Pentamethylene-7a-(nitroxymethyl)-3-oxoperhydroimidazo[1,5-c]oxazol-6-ium nitrate	−6166.4	−6159.0	−7.4	0.12
Diethylammonium diethyldithiocarbamate	−7639.6	−7650.0	10.4	−0.14
Diisobutylammonium diisobutyldithiocarbamate	−12,891.0	−12,868.4	−22.6	0.18
Diisopropylammonium diisopropyldithiocarbamate	−10,260.0	−10,252.6	−7.4	0.07
Dipropylammonium dipropyldithiocarbamate	−10,252.0	−10,271.7	19.7	−0.19
*N*,*N*-Dimethylglycine bisulfate	−2610.6	−2604.7	−5.9	0.23
*N*,*N*-Dimethylglycine methyl ester bisulfate	−3323.2	−3329.1	5.9	−0.18
*N*,*N*-Dimethylglycine methyl ester sulfate	−6765.2	−6790.2	25.0	−0.37
*N*,*N*-Dimethylglycine sulfate	−5371.6	−5346.6	−25.0	0.47
Tetraethylammonium nitrate	−5573.4	−5590.6	17.2	−0.31
Tetramethylammonium nitrate	−2960.5	−2958.5	−2.0	0.07
Tetra-n-butylammonium nitrate	−10,841.0	−10,818.4	−22.6	0.21

## Data Availability

The data presented in this study are available in Appendix A.

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
