# Peer review of "Revision and Extension of a Generally Applicable Group-Additivity Method for the Calculation of the Standard Heat of Combustion and Formation of Organic Molecules"

_molecules, 2021, doi:10.3390/molecules26206101_

Round 1

Reviewer 1 Report

The manuscript by Naef et al documents a cheminformatics approach to calculating the standard heats of combustion and formation of organic molecules. The method is based on using the known experimental values for ca. 5000 molecules to obtain energy values for different atom types and nearest neighbours. A simple summation of the corresponding terms, adjusted for their respective multiplicities, permits calculation of the standard heat of combustion. The corresponding standard enthalpy of formation is obtained upon subtraction of the standard enthalpies of combustion for the corresponding elements.  The paper builds on an earlier study (published in 2015) and extends the range of molecules included in the dataset, with special interest drawn to ionic liquids.

The paper is in general well written and easy to follow. There’s the occasional enthusiastic use of exclamation marks which in general should be frowned upon (sorry) for formal scientific writing. There are also various references to the application to ‘virtually any molecule’; again the authors need to reign this in a bit, as the method can only be applied to organic structures (as correctly reflected in the title). Inorganic molecules are still out of reach, although semi-empirical calculations such as PM7 have demonstrated good application here for the simulation of gas-phase heat of combustion enthalpies (see for instance Christopher, I. L., Michalchuk, A. A. L., Pulham, C. R., & Morrison, C. A. (2021). Towards Computational Screening for New Energetic Molecules: Calculation of Heat of Formation and Determination of Bond Strengths by Local Mode Analysis. Frontiers in chemistry9. https://doi.org/10.3389/fchem.2021.726357). As an aside, while acknowledging that access to sufficient reliable experimental enthalpies of formation for inorganic molecules may be out of reach, is there scope to train the authors cheminformatics approach against PM7 data?

The authors’ atom-group additivity method is based on the atom type and neighbour atoms only (as reflected in Table 4), but does this not leave longer, i.e. next neighbour, or next-next neighbour, interactions unaccounted for? In particular, the stabilisation effects of intramolecular hydrogen bonds would be missed, while the destabilisation effects of steric clashes would also be missed. I am struggling, in general, to see how molecular conformation is accounted for in their calculations; or rather while the author’s method certainly allows for the accurate determination of standard enthalpies of combustion and formation, we do not learn anything regarding molecular configuration from it – correct? In this regard it would be helpful to know if anything useful can be learned from the ca. 15% of molecules that were not fitted so well. Were the outlier data points simply rouge data points (as will undeniably be the case for some), or did some have longer range interactions that were not appropriately captured by the atom type/nearest-neighbour additive scheme?

A small (but extremely important) point – make sure to quote units in the Tables and Figures captions. The formatting of the Figures (e.g. axis legends and scales) could be greatly improved.  

Reviewer 2 Report

The paper of Naef and Acree is the prosecution of the ambitious project to evaluate the properties of a molecule, both physical and biological, by a computer approach based on the group additivity method.

In my opinion the project deserves great interest because it goes exactly in the direction now followed in the pharmaceutical research to reduce time consuming experiments and, in particular, experiments in which animals are used. We recall the great success obtained by evaluating log P, extensively used in QSAR, by the group additivity approach.

Notwithstanding no new achievement in respect to the previously published work is reached, it is noteworthy the incredible effort in collecting and analysing literature data to build up a complete database for the calculation of the standard heat of combustion and formation of organic molecules. The obtained results are statistically very significant and very useful for the scientific community.

Personally, I’m very interested in what happens in solution when the presence of the solvent is able to dramatically act on the conformation of the molecule and, in this way, to affect its biological properties. In these conditions the group contribution approach is not applicable for calculating some properties, but it is the same useful as probe to understand that something new is happening in solution.

In my opinion the paper deserves publication.
